# Lack of variations in the salamander chytrid fungus, *Batrachochytrium salamandrivorans*, at its alleged origin: Updating its Japanese distribution with new evidence

**David Lastra González** [1,2]*, **Kanto Nishikawa**[3], **Koshiro Eto**[4], **Shigeharu Terui**[5], **Ryo Kamimura**[6], **Nuria Viñuela Rodríguez**[7], **Natsuhiko Yoshikawa**[8], **Atsushi Tominaga**[1,6]

1 Department of Natural Sciences, Faculty of Education, University of the Ryukyus, Nishihara, Okinawa, Japan, 2 Department of Ecology, Faculty of Environmental Sciences, Czech University of Life Sciences Prague, Prague – Suchdol, Czech Republic, 3 Graduate School of Global Environmental Studies, Kyoto University, Yoshida Hon-machi, Sakyo, Japan, 4 Kitakyushu Museum of Natural History & Human History, Kitakyushu, Fukuoka, Japan, 5 Environment Grasp Promotion Network-PEG, Nonprofit Organization, Kushiro-shi, Hokkaido, Japan, 6 Graduate school of Engineering and Science, University of the Ryukyus, Nishihara, Okinawa, Japan, 7 Molecular Invertebrate Systematics and Ecology [MISE] Lab, Graduate School of Engineering and Science, University of the Ryukyus, Nishihara, Okinawa, Japan, 8 Department of Zoology, National Museum of Nature and Science, Tokyo, Tsukuba, Ibaraki, Japan

* g222001@edu.u-ryukyu.ac.jp

**Data Availability Statement:** All relevant data are within the manuscript and its Supporting

## Abstract

The chytrid fungus *Batrachochytrium salamandrivorans* [Bsal] is causing declines in the amphibian populations. After a decade of mapping the pathogen in Europe, where it is causing dramatic outbreaks, and North America, where its arrival would affect to the salamander's biodiversity hotspot, little is known about its current status in Asia, from presumably is native. Japan has several species considered as potential carriers, but no regulation is implemented against Bsal spreading. Previous Bsal known presence detected various cases on the Okinawa Island, southwestern Japan. Previous studies on its sister species, *B. dendrobatidis* presented a high genomic variation in this area and particularly on *Cynops ensicauda*. Here, we have done the largest monitoring to date in Japan on the *Cynops* genus, focusing on Okinawa Island and updating its distribution and providing more information to unravel the still unknown origin of Bsal. Interestingly, we have provided revealing facts about different detectability depending on the used molecular techniques and changes in its Japanese distribution. All in all, the Bsal presence in Japan, together with its low variability in the sequenced amplicons, and the lack of apparent mortalities, may indicate that this part of Asia has a high diversity of chytrids.

## Introduction

Chytridiomycosis, caused by the chytrid fungi *Batrachochytrium dendrobatidis* [Bd] and *B. salamandrivorans* [Bsal], is an amphibian disease that has resulted in population declines in many parts of the world [1]. Bsal has particularly affected salamanders and newts in Western

information files. My submission contains all raw data required to replicate the results of the study.

**Funding:** David Lastra González is supported by the JSPS Postdoctoral fellowships for Research in Japan The funders had no role in study design, data collection and analysis, decision to publish, or preparation of the manuscript.

**Competing interests:** The authors have declared that no competing interests exist.

Europe by infecting their skin and causing lethal lesions [2]. Bsal has affected both wild and captive populations, leading to a near extinction of the wild populations of fire salamander, *Salamandra salamandra*, in the Netherlands. It has caused a severe decline of the fire salamander in Belgium and Germany as well as an outbreak affecting *Triturus marmoratus* in NE Spain due to an illegal release of allochthonous species that affected local species as well [3–5].

The introduction of this fungal pathogen to Europe is primarily attributed to the pet salamander trade from East/Southeast Asia [6] and therefore, trade regulations have been established [7]. It is imperative to control the amphibian pet trade and conduct surveillance of the pathogen to prevent its spread to Bsal free regions [8].

Pursuing this need, the monitoring of Bsal in its the allegedly native distribution becomes primordial in order to know the ecological needs and the suitable areas where the human-mediated spreading could facilitate outbreaks caused by the arrival of Bsal. Primarily, Bsal-positive localities were found and thought to be part of the native distribution in South Korea, China, Vietnam, Taiwan, and Japan, suggesting an Asian origin of Bsal [6, 9].

Since the description of Bsal, monitoring efforts have been focused on Europe where massive screenings were carried out [3, 10]. Nevertheless, Asia has been subject to substantially less Bsal sampling, excluding Vietnam [6], China [11] and Taiwan [12].

Despite several localities with molecular confirmation for this chytrid fungus, since 2014 Bsal monitoring has not been performed in Japan. Thirteen specimens of four different species tested positive [9]. Regarding the other chytrid fungus species, Bd, Japan, and particularly the Ryukyu Islands, have been shown to have a high genetic diversity. As an example, [13] obtained 44 haplotypes of Bd ITS-DNA in Japan. Besides, the sword-tailed newts *Cynops ensicauda* from Okinawa Island, the main island of the Ryukyu Islands had both the highest infectious incidence and the greatest number of haplotypes. Remarkably, in both Okinawa Island and Amami Island the same species were Bsal positive [9].

Interestingly, to date, all the genomic data available belongs to the same isolate, and amplicons for Bsal positive samples were compared with these available sequences with not observed difference [9, 11]. Unfortunately, no further research has been done in any of the alleged Bsal native countries in order to assess for genetic variation, amplicon diversity, nor Bsal distribution.

Thus, in this research we present the biggest monitoring carried to date in Japan, where we have explored the genetic diversity of the Bsal positive animals in order to check if the same kind of variation as in Bd exist, which we have also included in our analysis, and we have checked thoroughly the alleged Bsal positives localities reported in the one and only study that included Japan, focusing on the hotspot of Bd, the Okinawa Island.

## Materials and methods

Between 2011 and 2023, a total of 1012 samples were collected from various sources for the purpose of detecting Bsal or unrelated sample collection. These samples were obtained from 98 sites, encompassing 21 amphibian species from Japan [See S1 Table]. We collected samples from 34 out of 43 prefectures of Japan, spanning across seven different islands. More than 50% of the samples belong to Japanese amphibians from the genus *Cynops* [*C. ensicauda* and *C. pyrrhogaster*]. Both were extensively sampled due to its known suitability as hosts for Bsal chytrid fungi [9].

Most of the samples consisted of skin swabs [Men-tip 1P1501, Nihon-Menbo Co.] collected using standard procedures for sampling amphibian chytrid fungi, while a small portion consisted of toe clippings [14]. When possible, and without compromising relevant data, swab

samples from the same locality and same amphibian species were pooled in order to reduce costs with a maximum number of four swabs pooled for one extraction [15].

Genomic DNA was extracted from both kind of samples, swabs and toe clips, using a slightly modified protocol [see 10] for the Qiagen DNeasy Blood and Tissue Kit [Qiagen, Hilden, Germany] and Bsal testing was conducted in the facilities of University of the Ryukyus. Two detection assays were utilized: one based on standard polymerase chain reaction [PCR] followed by electrophoresis [2] and the other on duplex Bd and Bsal quantitative PCR [16]. Positive or equivocal results on the standard PCR were reanalyzed using duplex quantitative PCR [qPCR]. Quantification standards were created from gBlocks [Integrated DNA Technologies Inc., Coralville, IA, USA] as Bsal and Bd standards. In qPCR tests, all analyses were performed in duplicate in Roche LightCycler 480 II [Roche Diagnostics, Tokyo, Japan], using the Roche 480 Probes Master mix [Roche Diagnostics, Tokyo, Japan]. A sample was considered positive only if both wells amplified, the increase in fluorescence showed a standard sigmoidal curve, and the Ct value was below 40. Samples with single well amplification were retested. In all analyses, we used the calculation of the 2nd derivative maximum, which is available in LightCycler 480 software. Subsequently, a subset of samples was sequenced by chain-terminating [Sanger] in Applied Biosystems 3130xl Genetic Analyzer [Applied Biosystems, Foster City, USA] targeting the 5.8S rRNA gene. This study was approved by the University of the Ryukyus Animal Experimentation Committee [approval number A2022053].

## Results

This study confirmed the presence of Bsal in Japan. Within the Japanese geographic range where it was detected, neither deaths nor animals with symptoms compatible with chytridiomycosis were observed.

Quantitative PCR [qPCR] analysis detected 17 samples of which at least one well became qPCR Bsal positive. According to the criteria mentioned in the Methods section, only four samples should be considered qPCR positive from two localities, but same species *Cynops ensicauda popei*, from Okinawa Island [see Table 1]. These four samples were checked again by qPCR and consistently showed a positive signal in both wells with permanent values around 1 GE load. In addition, we performed Sanger sequencing in all the doubtful samples, and we were successful in 13 of them. All sequenced samples (160 base pairs) showed a 100% identity with GenBank accession number KC762295. From the 13 sequences, 10 of them are from different localities, and two of them are coincident with previously published Bsal localities [9].

**Table 1. Results' summary of positive samples [N] of *Batrachochytrium dendrobatidis* [Bd] and *Batrachochytrium salamandrivorans* [Bsal] in Japan.**

| Island [N of localities with chytrid presence, N of samples] | N Bd+ Samples* | N Bsal + qPCR[†] | N Bsal + qPCR[‡] | N Bsal + Sequenced[§] | N Bd+ and Bsal+ |
|---|---|---|---|---|---|
| Okinawa [5,151] | 28 | 4 | 5 | 7 | [3+2] [¶] |
| Kyushu [3,168] | 2 | 0 | 0 | 4 | [0+1] [¶] |
| Honshu [1,263] | 0 | 0 | 0 | 1 | [0+0] [¶] |
| Hokkaido [2,74] | 2 | 0 | 0 | 1 | [0+1] [¶] |

* Bd + detected by qPCR following Blooi et al. 2013 [16].

[†] Bsal+ detected by qPCR as Blooi et al. 2013 [16] [less 40 Ct value, duplicate, sigmoidal curve].

[‡] Bsal+ detected by qPCR as Blooi et al. 2013 [16]. Following Spitzen-van der Sluijs et al. 2020, [18], a Bsal sample is considered positive if during qPCR analyses more qPCR replicates became positive than negative.

[§] Bsal+ sequenced by Sanger and 100% identical amplicon as Genbank reference accession number KC762295.

[¶] Bd and Bsal samples co-occurrence, [X+Y] being X the number of Bsal qPCR[†] positives that were also Bd+* and Y being the number of Bsal samples sequenced[§] and were also Bd+*.

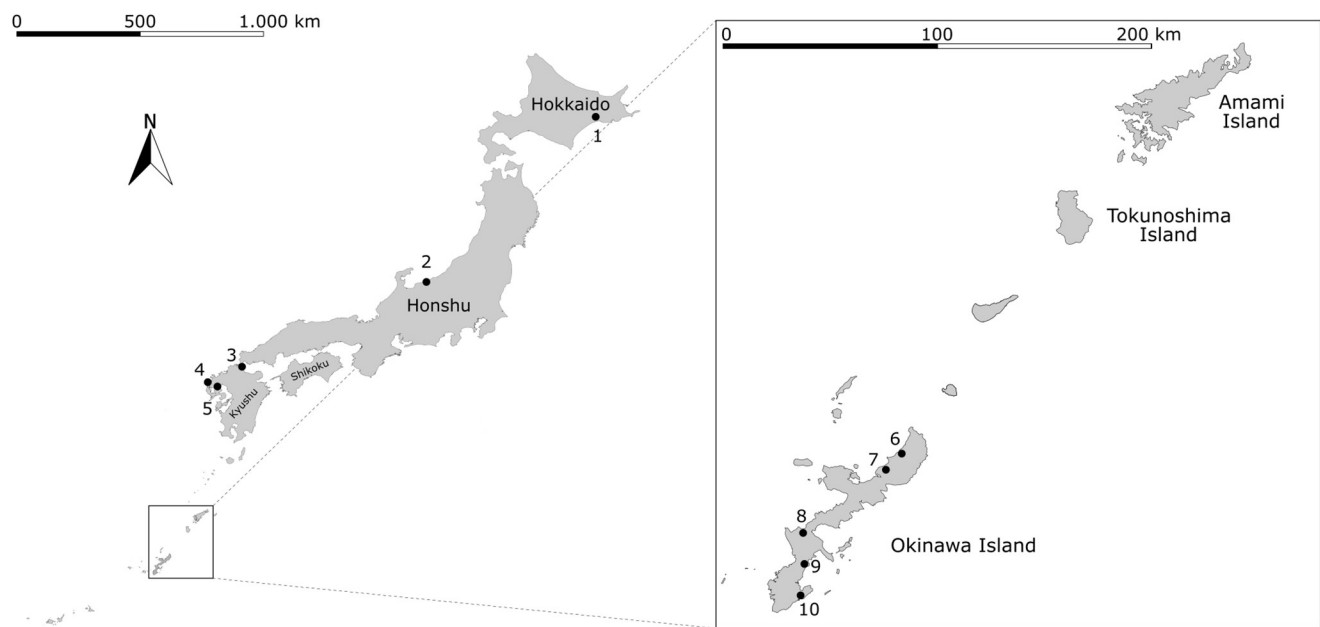

**Fig 1. Map of Bsal positive localities in Japan (Left) where Bsal DNA was obtained and had 100% match with Genbank accession number KC762295.** Detailed map of southern islands of Japan (Right). Number (N) 1 (*Hynobius retardatus*), 2 (*Cynops phyrrhogaster*), 3 (*Hynobius nebulosus*), 4 (*Cynops pyrrhogaster*), 5 (*Cynops pyrrhogaster*) 6–10 (*Cynops ensicauda*). Reprinted once the base map was created in QGIS (https://qgis.org/en/site/) with data obtained from the Geospatial Information Authority of Japan (Global Map Japan | GSI HOME PAGE <https://www.gsi.go.jp/kankyochiri/gm_japan_e.html>),] under a CC BY license, with permission from [the Geospatial Information Authority of Japan], original copyright [2024].

In the case of Bd, samples that were doubtful by standard PCR method were analyzed using the duplex Bd-Bsal qPCR. Here, Bd was detected in 32 samples [see Table 1]. One locality showed 55% of prevalence, but as in the rest of Bd positive samples, the GE load was always less than 10 GE.

In terms of distribution, we detected Bsal by sequencing amplicons on doubtful samples from Hokkaido, Honshu, and Kyushu islands, in addition to Okinawa Island. The Okinawan Bsal-positive localities are approximately 1000 km far to the nearest in Kyushu and monitoring in two small islands [Amamioshima and Tokunoshima] in between those with Bsal presence had not any animals with Bsal confirmation. [see Fig 1].

## Discussion

We have detected Bsal in Japan after 10 years of no information since the one and only study [9] containing data about the Bsal presence in Japan was published. In spite of the sampling effort for this work being four times larger, the epidemiologic situation seems analogous to [9]. The prevalence and the infection loads found, were similar to [9] and two out of the ten positive localities detected in this study correspond with already published positive localities. This evidence could suggest a long-term relationship among host and pathogen and most likely endemism of Bsal for at least Okinawa Island. The confirmed presence of the pathogen for more than a decade, the lack of individuals with high infection loads, and a low prevalence without detected outbreaks, respond better to an endemic situation in a relationship between host and pathogen. If we keep in mind the situation and how was the arrival of Bsal into the European urodelan populations, it does not show the same interactions of a novel pathogen into its potential hosts as it happened in Central Europe. On the other hand, one locality in

Okinawa Island, where Bsal was detected in 2014 [9], had no presence of newts despite several visits at different times of the year. Even when Bsal is known to be able to extirpate entire populations [3], we presumed that in this case it is related with human-mediated major habitat destruction of the surroundings, that left the area heavily altered and isolated.

Interestingly, our repeated analysis of samples by different methodologies presents an intriguing result. Bsal qPCR protocol has been under discussion and considered not enough to claim a sample for being Bsal positive [17]. Thus, in summary, the qPCR Bsal protocol was said to tend to create false positives. In contrast, we have observed that in 10 samples where we initially discarded the interpretation of qPCR results as Bsal positive, posterior standard PCR amplification in a different room where qPCR analyses were carried out, produced several amplicons with sequences 100% identical to the Bsal reference accession number KC762295.

As stated above, we have been very conservative in the establishment of criteria for assuming the positivity in our samples. If we would have followed the criteria of [18], for example, another five positives could be mentioned, meaning that from all repeated analysis more qPCR analyses were positive than negatives. On the other hand, we could have followed even a more conservative approach if we had confirmed the Bsal positive samples in two laboratories. Although qPCR is an established, reliable, and robust technique, and amplicons were identical to the published *B. salamandrivorans* sequence, the low-level infections in the Japanese amphibians preclude the use of an independent diagnostic technique as required by the World Organisation for Animal Health, hampering the possibility to exclude false positive results.

Another fact to point out is the Bsal presence in Hokkaido Island. In the past, two *Salamandrella keyserlingii* individuals out of four were found Bsal positive [9]. We increased the effort by ten times in this island and extended the monitoring to the only other caudate species found in the island, *Hynobius retardatus*, with one individual testing positive and being the first Bsal positive for this species. It is an intriguing situation that in a species with a high prevalence in the past [50% but a small sample size, see 9] Bsal was now not detected in this species. Hokkaido climate is known by its harsh winters, e.g., average temperature is below 0º C for 4 months, and this could have pushed Bsal to undetectable levels during the time of our monitoring. On the other hand, this detected/undetected situation has already occurred in other areas where previous known Bsal presence was detected and on different climate areas, e.g., Spain and Taiwan [see 12, 19]. However, its detection in *Hynobius retardatus* but not in *Salamandrella keyserlingii* is interesting, especially when this species was thought to be resistant to the fungus [9].

Unlike in its suggested sister species, Bd, we have not observed genetic variation in the Bsal sequenced amplicons in Okinawa [neither other part of Japan], a known hotspot for Bd haplotype variation [13]. A reasonable interpretation of this fact could be that the origin, or the evolutionary history, of both pathogens is different, meaning that Bsal presence in this area is at least more recent (or even could indicate an invasion event) than Bd or even artificially introduced. This suggestion is supported through evidence from sequences obtained from the rest of Japan share the same haplotype as those from Okinawa. Nonetheless, we are aware that the sequencing of a short fragment from the ITS region falls short for a firm interpretation, and additional methodologies such as Bsal strain typing would be recommended to infer sound conclusions.

Lastly, it is important to mention that, currently, Japanese conservation laws do not protect all amphibian species equally. For example, *Cynops pyrrhogaster* and *C. ensicauda popei*, in particular, are widely and frequently captured in the wild for commercial, research, and educational purposes. Furthermore, these species are easily and legally [as there is no regulation for this species] found in stores, private collections, or high schools [20]. It is crucial to implement

mitigation measures to prevent further spread of this deadly pathogen and protect amphibian populations as its presence in the international pet market is noteworthy [20, 21].

## Conclusions

Our study confirms the presence of *Batrachochytrium salamandrivorans* in Japan, suggesting that this fungus may have a longer history in this region in comparison other Asian countries where recurrent detection has not been achieved, and in addition, it is not causing obvious mass mortalities as it does in other parts of the world. The low pathogen load detected could be explained by the result of a long-term relationship between pathogen and host. However, similar relationships have been already found in Europe (e.g. *Ichtyosaura alpestris* [22]) even though Bsal does not show any sign of being native to Europe. Strong statements should be avoided with the current available information. The found prevalence existing in our monitoring, 1.6%, is slightly lower (but not significantly lower) than in other studies in Bsal presumably native countries [e.g., China, 2.9% and Vietnam, 2.9%, see 6, 11]. Until further phylogenetic analyses confirm its origin, the Bsal presence in Japanese territory may have been introduced by humans, being endemic to Southeast/East Asia or even circulate undetected in this geographic range without causing deaths as well. These results should raise awareness among the research and conservation communities and prompt urgent action to identify regions with early emergence of the disease and implement mitigation measures to prevent further spread of this deadly pathogen.

## Supporting information

**S1 Table. Detailed information about amphibian collected samples.** *Bsal+ detected by Blooi et al. 2013 (less 40 Ct value, duplicate, sigmoidal curve). †Bsal+ detected by Blooi et al. 2013 (more qPCR positives than negatives as Spitzen-van der Sluijs et al. 2020). ‡Bsal + sequenced by Sanger and 100% identical amplicon as Genbank reference accession number KC762295.
(DOCX)

## Acknowledgments

We would like to thank to Jiro Kawahara, Asaki Hentona, Daichi Shingaki, Kousuke Uchiwa, Sakura Nishijima and Ichiro Yoshida for their fieldwork assistance and to Vojtech Baláž, Matthew J.Gray and Edward Davis Carter for their laboratory assistance. Board of Education of Kushiro city kindly permitted collecting swab from *Salamandrella keyserlingii* [Reference No. 6, 2023 to S. Terui]. The Ministry of Environment of Japan permits us to collect samples from *Onychodactylus tsukubaensis* [Reference number N. 2303173 to N. Yoshikawa] and provided samples of *Hynobius abei*. The Agency for Cultural Affairs of Japan permit us to collect samples from *Andrias japonicus* [Reference number N. 23-4-1944 to K. Nishikawa]. We also thank the Ministry of the Environment of Japan [2301101 to A. Tominaga] and Kagoshima Prefectural Board of Education [reference No. 5244, 2023 to A. Tominaga] and the Okinawa Prefectural Board of Education [reference No. 77, 2022 to A. Tominaga] for providing permissions for collecting samples for *Echinotriton andersoni* and *Echinotriton raffaellii*.

## Author Contributions

**Conceptualization:** David Lastra González.

**Formal analysis:** David Lastra González, Atsushi Tominaga.

**Funding acquisition:** David Lastra González.

**Investigation:** David Lastra González.

**Methodology:** David Lastra González.

**Resources:** David Lastra González, Kanto Nishikawa, Koshiro Eto, Shigeharu Terui, Natsu-hiko Yoshikawa, Atsushi Tominaga.

**Software:** Ryo Kamimura, Nuria Viñuela Rodríguez, Atsushi Tominaga.

**Supervision:** David Lastra González, Atsushi Tominaga.

**Visualization:** Nuria Viñuela Rodríguez.

**Writing – original draft:** David Lastra González, Atsushi Tominaga.

**Writing – review & editing:** David Lastra González, Kanto Nishikawa, Koshiro Eto, Shigeharu Terui, Ryo Kamimura, Nuria Viñuela Rodríguez, Natsuhiko Yoshikawa, Atsushi Tominaga.

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
