## [Decision Letter · Decision Letter 0]

16 Jan 2024

PONE-D-23-38081Disentangling the origin of the salamander killer fungus, Batrachochytrium salamandrivorans: Updating its Japanese distribution with new evidencePLOS ONE

Dear Dr. Lastra González,

Thank you for submitting your manuscript to PLOS ONE. After careful consideration, we feel that it has merit but does not fully meet PLOS ONE’s publication criteria as it currently stands. Therefore, we invite you to submit a revised version of the manuscript that addresses the points raised during the review process.

We look forward to receiving your revised manuscript.

Kind regards,

Tzen-Yuh Chiang

Academic Editor

PLOS ONE

 [David Lastra González is supported by the JSPS Postdoctoral fellowships for Research in Japan].  

4. Ethics statement appears in the Methods section of the manuscript AND at the end of the manuscript:

Your ethics statement should only appear in the Methods section of your manuscript. If your ethics statement is written in any section besides the Methods, please delete it from any other section. 

Reviewers' comments:

Reviewer's Responses to Questions

**Comments to the Author**

1. Is the manuscript technically sound, and do the data support the conclusions?

Reviewer #1: Yes

Reviewer #2: Partly

2. Has the statistical analysis been performed appropriately and rigorously? 

Reviewer #1: Yes

Reviewer #2: N/A

3. Have the authors made all data underlying the findings in their manuscript fully available?

Reviewer #1: Yes

Reviewer #2: Yes

4. Is the manuscript presented in an intelligible fashion and written in standard English?

Reviewer #1: No

Reviewer #2: No

5. Review Comments to the Author

Reviewer #1: This is an interesting study in the field of Bsal research, which is a dangerous amphibian pathogen in certain parts of the world, so far especially Europe. As Asia is considered the origin of Bsal, this study is an important contribution bringing to light novel knowledge. The “story” is sound, the introduction leads to the purpose of the paper and the methods are standard methods. Results are well described and convincing. The discussion refers to the significant points and emphasizes the importance of the finding in a critical view.

I like the paper and will be happy to see it published in PLoS ONE. I made some comments, see below, which might help the authors to improve their paper.

However, before acceptance, in many parts, the language is awkward. Maybe look for a native speaker to make corrections.

Title, line 1: I find the title a bit tough, as Bsal is not always a killer, especially not in Japan. I recommend to change the title to simply use the term “salamander chytrid fungus”

Introduction, line 53: Maybe rewrite as “in many parts of the world” instead of worldwide.

Line 59-63: True but a bit out of focus of the paper, I suggest to be more brief here.

Line 66 f: Emphasize here that this study and others (e.g. 6, 11, 12) have suggested already that Bsal is of Asian origin.

Line 84f. You should state here that also Bd was in your focus, as you studied both (mentioned in line 109).

Methods, line 94: delete “different” – 22 species, by definition, are different.

Results, line 128f.: Although important this I part of the method and was mentioned there already (line 114f.). Do not repeat method here. Move details to method section and rewrite.

Line 145f.: Add the reference number to the citation of Bloii et al. and Spitzen-van der Sluijs et al.

Discussion, line 164: “This evidence suggests a long-term relationship among host and pathogen…” Explain this a bit better.

Line 171f: “Bsal qPCR protocol was under discussion and considered not enough to claim a sample for being Bsal positive [17]. Thus, in summary. the qPCR Bsal protocol was said to tend to create false positives.” This likewise needs a better explanation for the reader.

Line 177: Yes, I agree you followed a “conservative” approach but still samples were not studied in different labs, which sometimes is considered a critical aspect, as well. Please consider to include this into the discussion here.

Line 193: Say “suggested sister species”.

Reviewer #2: Summary

This paper presents data on the prevalence of the fungal pathogen Bsal (and its congener Bd) in samples collected from multiple Japanese islands (as well as 2 samples from Myanmar). The authors find a low prevalence of Bsal-positive samples (comparable to that in other surveys in East and Southeast Asia), but some inconsistencies in detection between different diagnostic methods. In my opinion, the most useful contributions from this paper are the establishment of new Bsal localities in East Asia, some associated sequence data, and the detection in a species with no previous record of Bsal.

Major concerns

The paper is in need of substantial English-language editing throughout. The meaning of the text is often clear, but there are key passages where it is not, and key terms are not clearly defined. For example, it is hard to tell what species is being referred to in the description of the outbreak in NE Spain (Line 57)

There are a few places where I think the conclusions and interpretations don’t seem to be in line with the data. For example, the suggestion that “this part of Asia is the chytrid hotspot” in the abstract doesn’t seem consistent with their results and interpretations, but it is hard to tell if this is a language issue or an interpretation problem. Also, I think the authors should be more cautious about interpretations related to long-term endemism based on the limited data they provide.

The organization of the Introduction could be much clearer, including both the flow of ideas within paragraphs and the relationships between paragraphs. For example, there is a rather abrupt transition from Bsal to Bd on line 74, and there is a series of very short paragraphs between lines 59 and 71 that seem too small to stand on their own and don’t really flow together very well.

It would be good to have more discussion of why they got disparate results using different detection methods.

Specific comments

Line 54: Are the lesions themselves lethal? Or are they a symptom associated with severe disease, which causes death through other mechanisms?

Lines 56-58: Rewrite sentence for clarity.

Lines 59-63: Good points, but are not very clearly connected to the surrounding material in the Introduction, and may be less relevant for this particular study.

Lines 80-82: Meaning not clear

Line 95: The Introduction focuses on Japan, so I was a little bit surprised to see that there were samples from Myanmar as well. I didn’t see these mentioned anywhere else in the paper.

Line 117: Unclear what the 2nd derivative maximum was used for.

Table 1: Would be good to have the total number of samples per island listed somewhere

Line 162: I’m not sure why sampling effort would be related to infection load

Lines 164-165: Not clear how the data suggests long-term relationship and endemism.

Lines 170-180: The discussion of different diagnostic techniques seems useful. Would be good to flesh this out more, including adding more information about the differences in sensitivity between the assays used here and other approaches.

Line 186-188: Do we know that harsh winters decrease probability of detecting Bsal? Presumably the hosts are in habitats that are not quite so cold?

Lines 191-192: I agree that it is interesting and useful to expand the list of species on which Bsal has been detected.

Line 191-205: nice paragraph

Lines 210-211: I don’t think the current paper has enough data to support this point, and I’m not sure I agree with the logic (eg, there can be species in the introduced range that also tend to have low Bsal loads).

Lines 214-216: This sentence doesn’t seem to fit very well

Map figure: would be useful if this showed all sampled localities (not just the Bsal-positive ones)

6. PLOS authors have the option to publish the peer review history of their article (what does this mean?). If published, this will include your full peer review and any attached files.

Reviewer #1: No

Reviewer #2: No

---

## [Author Response · Author response to Decision Letter 0]

24 Jan 2024

Dear Editor and Reviewers,

All your comments have been carefully implemented.

Thank you for your time and your comments.

Kind regards,

David Lastra González

---

## [Decision Letter · Decision Letter 1]

5 Mar 2024

PONE-D-23-38081R1Disentangling the origin of the salamander chytrid fungus, Batrachochytrium salamandrivorans: Updating its Japanese distribution with new evidencePLOS ONE

Dear Dr. Lastra González,

Thank you for submitting your manuscript to PLOS ONE. After careful consideration, we feel that it has merit but does not fully meet PLOS ONE’s publication criteria as it currently stands. Therefore, we invite you to submit a revised version of the manuscript that addresses the points raised during the review process.

We look forward to receiving your revised manuscript.

Kind regards,

Tzen-Yuh Chiang

Academic Editor

PLOS ONE

Reviewers' comments:

Reviewer's Responses to Questions

**Comments to the Author**

1. If the authors have adequately addressed your comments raised in a previous round of review and you feel that this manuscript is now acceptable for publication, you may indicate that here to bypass the “Comments to the Author” section, enter your conflict of interest statement in the “Confidential to Editor” section, and submit your "Accept" recommendation.

Reviewer #1: All comments have been addressed

Reviewer #3: (No Response)

2. Is the manuscript technically sound, and do the data support the conclusions?

Reviewer #1: Yes

Reviewer #3: No

3. Has the statistical analysis been performed appropriately and rigorously? 

Reviewer #1: Yes

Reviewer #3: N/A

4. Have the authors made all data underlying the findings in their manuscript fully available?

Reviewer #1: Yes

Reviewer #3: Yes

5. Is the manuscript presented in an intelligible fashion and written in standard English?

Reviewer #1: Yes

Reviewer #3: Yes

6. Review Comments to the Author

Reviewer #1: (No Response)

Reviewer #3: The authors describe an extensive sampling of amphibians throughout Japan for chytrid fungi. Results are largely confirmatory of previous studies (eg Goka et al., 2009; Martel et al., 2014) but are sufficiently interesting to warrant publication. While in general, the study is fine, I would recommend addressing several issues:

Major concerns:

I only have one major concern and that is that sequencing a short (how much bp? Please mention) part of the ITS of the chytrid fungus B. salamandrivorans is not sufficient to make any inference about diversity. I am not sure whether I understood correctly, but sequences would all be 100% identical (and identical to the available sequence). The authors really should refrain from drawing any conclusions with regard to diversity / phylogeny based on these results. At most, the sequencing confirms that a short B. salamandrivorans ITS sequence is present in their samples. Hence, conclusions with regard to “origin”, “hotspot” or any inference with regard to epidemiology (eg recent colonization) must not be drawn. Such statements should be omitted from title, abstract and discussion, unless the authors would have performed additional analyses that do warrant conclusions with regard to B. salamandrivorans strain typing (eg strain isolation and sequencing).

Minor concerns:

1) There is an odd mentioning of two samples from Myanmar. It is difficult to understand how this fits the story. Suggestion to omit from this paper (does not really add anything useful). Also, if the authors would prefer to leave these samples in, please provide all permits (incl Nagoya protocol) from Myanmar to demonstrate legal original (+ correct the name in S1 to Tylototriton). However, there does not appear to be any sound, scientific rationale to include these two samples.

2) Results need revision; the table 1 is confusing and at the same time does not provide sufficient information. For example, the difference between the second and third column is unclear. The total number of sampled localities per island + the number of samples (pooled or not) examined should be summarized here. The same for the legend of S Table, which is equally difficult to understand. E.g. the legend states “detected by Blooi et al…”, where I guess the authors mean “detected by qPCR (according to Blooi et al.).

3) It is difficult to track any data that refer to prevalence or infection load. The authors used pooled and simple samples. This is ok, but pooling samples requires any conclusion to prevalence and infection loads to be nuanced (eg prevalence then is at pool level).

4) There is one issue where I would love to see an additional sentence and that is the potential of false positives (which includes the possibility of another organism with the same (short) ITS sequence). According to WOAH, diagnosis of B. salamandrivorans infection requires the use of two independent techniques (eg culturing or histopathology). Sequencing amplicons from a positive PCR is not considered an independent technique. I would like to stress this is not meant to disregard the authors’ results. I fully realize it is very difficult, if not impossible, to use an additional technique (culturing, histopathology), especially in low level infected populations. However, if not performed, it is impossible to discard the possibility of false positives (which has been the case in Europe, see eg Thomas et al. and which remains unexplained). I would therefore recommend to add a sentence along these lines: “Although qPCR is an established, reliable and robust technique and amplicons were identical to the published B. salamandrivorans sequence, the low level infections in the Japanese amphibians preclude the use of an independent diagnostic technique as required by the WOAH, hampering the possibility to exclude false positive results.”

7. PLOS authors have the option to publish the peer review history of their article (what does this mean?). If published, this will include your full peer review and any attached files.

Reviewer #1: No

Reviewer #3: No

---

## [Author Response · Author response to Decision Letter 1]

14 Mar 2024

All your comments has been revised and answered in the "Response to reviewers" file. Have a nice day. David

---

## [Decision Letter · Decision Letter 2]

1 Apr 2024

PONE-D-23-38081R2Disentangling the origin of the salamander chytrid fungus, Batrachochytrium salamandrivorans: Updating its Japanese distribution with new evidencePLOS ONE

Dear Dr. Lastra González,

Thank you for submitting your manuscript to PLOS ONE. After careful consideration, we feel that it has merit but does not fully meet PLOS ONE’s publication criteria as it currently stands. Therefore, we invite you to submit a revised version of the manuscript that addresses the points raised during the review process.

We look forward to receiving your revised manuscript.

Kind regards,

Tzen-Yuh Chiang

Academic Editor

PLOS ONE

Reviewers' comments:

Reviewer's Responses to Questions

**Comments to the Author**

1. If the authors have adequately addressed your comments raised in a previous round of review and you feel that this manuscript is now acceptable for publication, you may indicate that here to bypass the “Comments to the Author” section, enter your conflict of interest statement in the “Confidential to Editor” section, and submit your "Accept" recommendation.

Reviewer #1: All comments have been addressed

Reviewer #3: All comments have been addressed

Reviewer #4: (No Response)

2. Is the manuscript technically sound, and do the data support the conclusions?

Reviewer #1: Yes

Reviewer #3: Yes

Reviewer #4: No

3. Has the statistical analysis been performed appropriately and rigorously? 

Reviewer #1: Yes

Reviewer #3: N/A

Reviewer #4: N/A

4. Have the authors made all data underlying the findings in their manuscript fully available?

Reviewer #1: Yes

Reviewer #3: Yes

Reviewer #4: Yes

5. Is the manuscript presented in an intelligible fashion and written in standard English?

Reviewer #1: Yes

Reviewer #3: Yes

Reviewer #4: Yes

6. Review Comments to the Author

Reviewer #1: (No Response)

Reviewer #3: Concerns have been addressed sufficiently. I commend the authors for their extensive study and have no further concerns.

Reviewer #4: The title is inaccurate. Their is no phylogenetic analysis in this paper or comparative measures of genetic diversity. A more appropriate title could be: 'An updated distribution of Batrachochytrium salamandrivorans in Japan’.

L41. It's prevalence - not incidence. And it is not high - it is actually very low compared to outbreak areas in Europe.

L48. 'Low variability' is a relative term. What are you comparing against? Epidemiologically speaking, low diversity indicates an invasion event (limited temporal time allowed for genetic diversity to accrue). So your data more accurately represents a recent invasion.

L108. What is the evidence that the metabarcoding marker that you have chosen has any phylogenetic resolution? O'Hanlon et al. has shown very clearly (see Supp Info tanglegram plots) that ITS-2 has no phylogenetic resolution for Bd.

L131. Sanger sequencing of what locus?

L223. This is a hypothesis. For instance in Europe only Salamandra is highly susceptible. Other species of newt tolerate infection or are resistant to infection.

7. PLOS authors have the option to publish the peer review history of their article (what does this mean?). If published, this will include your full peer review and any attached files.

Reviewer #1: No

Reviewer #3: No

Reviewer #4: No

---

## [Author Response · Author response to Decision Letter 2]

2 Apr 2024

All comments of the last reviewer (4th Reviewer) have been addressed on the Response to Reviewers file

---

## [Decision Letter · Decision Letter 3]

9 May 2024

PONE-D-23-38081R3Disentangling the origin of the salamander chytrid fungus, Batrachochytrium salamandrivorans: Updating its Japanese distribution with new evidencePLOS ONE

Dear Dr. Lastra González,

Thank you for submitting your manuscript to PLOS ONE. After careful consideration, we feel that it has merit but does not fully meet PLOS ONE’s publication criteria as it currently stands. Therefore, we invite you to submit a revised version of the manuscript that addresses the points raised during the review process.

We look forward to receiving your revised manuscript.

Kind regards,

Tzen-Yuh Chiang

Academic Editor

PLOS ONE

Reviewers' comments:

Reviewer's Responses to Questions

**Comments to the Author**

1. If the authors have adequately addressed your comments raised in a previous round of review and you feel that this manuscript is now acceptable for publication, you may indicate that here to bypass the “Comments to the Author” section, enter your conflict of interest statement in the “Confidential to Editor” section, and submit your "Accept" recommendation.

Reviewer #1: All comments have been addressed

Reviewer #4: (No Response)

2. Is the manuscript technically sound, and do the data support the conclusions?

Reviewer #1: Yes

Reviewer #4: No

3. Has the statistical analysis been performed appropriately and rigorously? 

Reviewer #1: Yes

Reviewer #4: No

4. Have the authors made all data underlying the findings in their manuscript fully available?

Reviewer #1: Yes

Reviewer #4: Yes

5. Is the manuscript presented in an intelligible fashion and written in standard English?

Reviewer #1: Yes

Reviewer #4: No

6. Review Comments to the Author

Reviewer #1: (No Response)

Reviewer #4: I still think the title is inaccurate. Even if your aim is to contribute to the elucidation of the origin of Bsal, your data do not say anything about this, so to include ‘“Disentangling the origin” is confusing.

As you argue in your replay, the term of incidence is confusing, so please replace the the term ‘incidence’ in the abstract or use another term to avoid misunderstandings.

As you agree in your replay, a low diversity usually indicates an invasion event. Therefore, please discuss this point more in detail in the discussion.

Regarding the use of ITS-2, again, your data do not tell us anything about the origin of Bsal and are only valuable for the distribution of Bsal.

Regarding your conclusions, again I think your data are not informative to say ‘this fungus has a long history in this region’.

Additionally, the sentence ‘The found prevalence existing in our monitoring, 1.6%, is lower than in other studies in presumably Bsal native countries’ is incorrect because a proportion of 11/650 (your study) is not significantly lower than 33/1143 (China) nor 17/583 (Vietnam; Z=1.572, p=0.1160; Z=1.44, p=0.1499; respectively).

7. PLOS authors have the option to publish the peer review history of their article (what does this mean?). If published, this will include your full peer review and any attached files.

Reviewer #1: **Yes: **Stefan Lötters

Reviewer #4: No

---

## [Author Response · Author response to Decision Letter 3]

16 May 2024

6. Review Comments to the Author

Please use the space provided to explain your answers to the questions above. You may also include additional comments for the author, including concerns about dual publication, research ethics, or publication ethics. (Please upload your review as an attachment if it exceeds 20,000 characters).

- Reviewer #1: (No Response)

- Reviewer #4: I still think the title is inaccurate. Even if your aim is to contribute to the elucidation of the origin of Bsal, your data do not say anything about this, so to include ‘“Disentangling the origin” is confusing.

- DLG: The title has been modified as you suggested: Lack of variations in the salamander chytrid fungus, Batrachochytrium salamandrivorans, at its alleged origin: Updating its Japanese distribution with new evidence.

- As you argue in your replay, the term of incidence is confusing, so please replace the the term ‘incidence’ in the abstract or use another term to avoid misunderstandings.

DLG: In order to avoid this, the term high incidence has been changed by various cases.

- As you agree in your replay, a low diversity usually indicates an invasion event. Therefore, please discuss this point more in detail in the discussion.

- DLG: We have added your suggestion (low diversity could indicate an invasion event). It has been done in the area where we had discussed this matter (Line 212 in the current version).

- Regarding the use of ITS-2, again, your data do not tell us anything about the origin of Bsal and are only valuable for the distribution of Bsal.

- DLG: We are trying to show here the contrast between Bd and Bsal. We know that Bd has a high variability when sequenced in Okinawa (see for more Goka et al 2009, 2021). On the other hand, we have not detected any variability in the sequences obtained from Bsal in Okinawa. This situation and lack of variability among Bsal sequences could be interpreted as a difference in the time that both pathogens have been present in Okinawa, and I hope that you interpret our data in the same way.

- Regarding your conclusions, again I think your data are not informative to say ‘this fungus has a long history in this region’.

Additionally, the sentence ‘The found prevalence existing in our monitoring, 1.6%, is lower than in other studies in presumably Bsal native countries’ is incorrect because a proportion of 11/650 (your study) is not significantly lower than 33/1143 (China) nor 17/583 (Vietnam; Z=1.572, p=0.1160; Z=1.44, p=0.1499; respectively).

- DLG: We have revised the Conclusion to address your points. Our results do not suggest, as you say, that this fungus has a long history in the area, but we have at least stated that it is highly compatible with that idea. For example, If you check all the studies that have been done about Bsal in Asia, (China, Vietnam, and Taiwan), you may notice that after the first detection carried out in these countries Bsal was not detected in the same localities or even to confirm the Bsal presence in the country was not possible again. However if you have any problem with this term, I will be more specific and being less assertive, rephrasing the sentence in this manner: “Our study confirms the presence of Batrachochytrium salamandrivorans in Japan, suggesting that this fungus may have a longer history in this region in comparison other Asian countries where recurrent detection has not been achieved, and in addition, it is not causing obvious mass mortalities as it does in other parts of the world”.

About the second part, the numbers of the prevalence, we have not mentioned any statistical term along the text. Your calculations are correct, but we are talking about the raw numbers, and we were not including the word significant, so our information about the prevalence is correct. In order to avoid any misleading information for the reader we have included between brackets “(But not significantly lower”) and “slightly” lower in the part that you have mentioned. Please see the text, line 238.

---

## [Decision Letter · Decision Letter 4]

28 May 2024

Lack of variations in the salamander chytrid fungus, Batrachochytrium salamandrivorans, at its alleged origin: Updating its Japanese distribution with new evidence.

PONE-D-23-38081R4

Dear Dr. González,

We’re pleased to inform you that your manuscript has been judged scientifically suitable for publication and will be formally accepted for publication once it meets all outstanding technical requirements.

Kind regards,

Tzen-Yuh Chiang

Academic Editor

PLOS ONE

Additional Editor Comments (optional):

Reviewers' comments:

Reviewer's Responses to Questions

**Comments to the Author**

1. If the authors have adequately addressed your comments raised in a previous round of review and you feel that this manuscript is now acceptable for publication, you may indicate that here to bypass the “Comments to the Author” section, enter your conflict of interest statement in the “Confidential to Editor” section, and submit your "Accept" recommendation.

Reviewer #1: All comments have been addressed

2. Is the manuscript technically sound, and do the data support the conclusions?

Reviewer #1: Yes

3. Has the statistical analysis been performed appropriately and rigorously? 

Reviewer #1: Yes

4. Have the authors made all data underlying the findings in their manuscript fully available?

Reviewer #1: Yes

5. Is the manuscript presented in an intelligible fashion and written in standard English?

Reviewer #1: Yes

6. Review Comments to the Author

Reviewer #1: Regarding the MS by Lastra et al. "Lack of variations in the salamander chytrid fungus, Batrachochytrium salamandrivorans, at its alleged origin: Updating its Japanese distribution with new evidence", I look forward to seeing it published.

7. PLOS authors have the option to publish the peer review history of their article (what does this mean?). If published, this will include your full peer review and any attached files.

Reviewer #1: No

---

## [Editor Report · Acceptance letter]

3 Jun 2024

PONE-D-23-38081R4 

PLOS ONE

Dear Dr. Lastra González, 

I'm pleased to inform you that your manuscript has been deemed suitable for publication in PLOS ONE. Congratulations! Your manuscript is now being handed over to our production team.

Kind regards, 

on behalf of

Dr. Tzen-Yuh Chiang 

Academic Editor

PLOS ONE